# Nodal Elective Volume Selection and Definition during Radiation Therapy for Early Stage (T1–T2 N0 M0) Perianal Squamous Cell Carcinoma: A Narrative Clinical Review and Critical Appraisal

**DOI:** 10.3390/cancers15245833

**Published:** 2023-12-14

**Authors:** Lavinia Spinelli, Stefania Martini, Salvatore Dario Solla, Riccardo Vigna Taglianti, Francesco Olivero, Luca Gianello, Alessia Reali, Anna Maria Merlotti, Pierfrancesco Franco

**Affiliations:** 1Radiation Oncology Department, Santa Croce and Carle Hospital, 12100 Cuneo, Italy; spinelli.l@ospedale.cuneo.it (L.S.); martini.st@ospedale.cuneo.it (S.M.); solla.s@ospedale.cuneo.it (S.D.S.); vigna.r@ospedale.cuneo.it (R.V.T.); olivero.franc@ospedale.cuneo.it (F.O.); gianello.l@ospedale.cuneo.it (L.G.); merlotti.a@ospedale.cuneo.it (A.M.M.); 2Radiation Oncology Department, Michele and Pietro Ferrero Hospital, 12060 Verduno, Italy; areali@asl.cn2.it; 3Department of Translational Medicine (DIMET), University of Eastern Piedmont, 28100 Novara, Italy; 4Department of Radiation Oncology, ‘Maggiore della Carità’ University Hospital, 28100 Novara, Italy

**Keywords:** perianal squamous cell carcinoma, radiotherapy, treatment volumes, contouring, anal cancer, narrative review

## Abstract

**Simple Summary:**

Early-stage (T1–T2 N0 M0) true perianal tumors are very uncommon, and the scientific literature is scant. Based on common features with anal canal carcinomas (aCCs), perianal skin cancers and aCCs are included in the same tumor classification and treated similarly. In fact, contouring radiation therapy guidelines do not differentiate between the two subsites. However, anal canal tumors and perianal skin cancers have different lymphatic drainage patterns. Modulation of radiotherapy treatment volumes can be considered for the latter. We performed a literature review to analyze the sites at higher risk of microscopic spread in patients with early-stage perianal cancer to tailor the selection of radiation therapy elective volumes.

**Abstract:**

Distinction between anal canal and perianal squamous cell carcinomas (pSCCs) is essential, as these two subgroups have different anatomical, histological, and lymphatic drainage features. Early-stage true perianal tumors are very uncommon and have been rarely included in clinical trials. Perianal skin cancers and aCCs are included in the same tumor classification, even though they have different lymphatic drainage features. Furthermore, pSCCs are treated similarly to carcinomas originating from the anal canal. Radiation therapy (RT) is an essential treatment for anal canal tumors. Guidelines do not differentiate between treatment volumes for perianal tumors and anal cancers. So far, in pSCC, no study has considered modulating treatment volume selection according to the stage of the disease. We conducted a narrative literature review to describe the sites at higher risk for microscopic disease in patients with early-stage perianal cancers (T1–T2 N0 M0) to propose a well-thought selection of RT elective volumes.

## 1. Introduction

The anal region is divided into the anal canal, which extends for 3–4 cm from the superior margin of the internal anal sphincter to the anal verge, and the skin surrounding the anal verge and extending 5–6 cm, radially. This distinction is important with respect to lymphatic drainage and therapeutic options.

Perianal squamous cell carcinomas (pSCCs) are rare. They are fivefold less common than anal canal neoplasms [1,2]. Most are well to moderately differentiated keratinizing SCCs. Other tumors include Bowen disease (SCC in situ), basal cell carcinoma, verrucous carcinoma, melanoma, and Paget disease (adenocarcinoma in situ).

Multiple risk factors have been reported. As with cervical cancer, perianal and anal canal cancer are strongly associated with infection with human papillomavirus (HPV), especially genotypes 16 and 18 [3]. Additional risk factors include a history of receptive anal intercourse, sexually transmitted diseases, more than ten sexual partners throughout life, a history of other anogenital cancer (cervical, vulvar, or vaginal), and immunosuppression after solid-organ transplantation [4].

Perianal tumors are generally ulcerated, hard when palpated, and raised with rolled, everted edges. Most patients present in the seventh to eighth decades of life with an approximately equal male-to-female ratio [5]. Symptoms are nonspecific and include bleeding, pain, exudation, and pruritus. Due to nonspecific complaints, the diagnosis is often delayed.

Up to the seventh edition of the American Joint Committee on Cancer (AJCC) cancer staging manual, cancers of the hair-bearing perianal skin were classified and staged as skin tumors [6]. The anal canal and margin are often both clinically involved, and sphincter preservation is crucial. Hence, perianal lesions have progressively been treated similarly to carcinomas originating from the anal canal. Indeed, the eighth and ninth editions of the AJCC staging manuals include ‘Anal canal and perianal skin’ in the same tumor classification. The following definitions were introduced: anal canal lesions are lesions that cannot be visualized at all or are incompletely visualized with gentle traction placed on the buttocks; in contrast, true perianal lesions are completely visible and fall within a 5 cm radius of the anal opening; finally, skin lesions fall outside the 5 cm radius of the anal opening (Figure 1).

In the TNM classification for anal cancer, primary tumor (T) stages are primarily based on tumor size, except for the T4 stage, which is based on the invasion of adjacent organs. This is different from most gastrointestinal malignancies, which are staged according to the depth of invasion. The definition of regional lymph nodes has also changed from the seventh to the eighth edition. The latter adds external iliac lymph nodes to the already recognized perirectal, internal iliac, and inguinal lymph nodes. Whereas metastases in regional lymph node(s) (N) are categorized as N1, N2, and N3 in the TNM seventh edition, they are all categorized as N1 in the TNM eighth edition. Thus, N2 and N3 are abolished [7]. Table 1 below summarizes the most recent TNM classifications for anal cancer.

The location of a tumor is essential for radiation therapy (RT) treatment planning, as the lymphatic drainage of the anatomical anal canal differs from that of the perianal skin. Perianal skin drains mainly to the inguinal lymph nodes, while the distal anal canal, together with the mid-canal and proximal canal, drain to the internal iliac and superior hemorrhoidal lymph nodes [8].

Tumors arising from the anal region tend to spread locally to the surrounding soft tissues and regional lymph nodes due to a strong tropism for lymphatic vessels. Anal canal tumors can involve pelvic nodes, starting from hemorrhoidal and continuing to external iliac, internal iliac, and obturator nodes. The involvement of inguinal nodes is quite characteristic. Overall, the more distal the lesion and the closer to skin, the more likely inguinal involvement is.

Perianal squamous cell carcinomas usually have a more favorable outcome than anal canal cancers. The main prognostic factors are T-stage and lymph node involvement at diagnosis [9,10,11].

Metastases to the inguinal nodes occur in 15 to 25% of patients and are related to the size and differentiation of the primary tumor. Tumors sized 2 to 5 cm and ≥5 cm have 24% and 67% rates of nodal involvement at presentation, respectively [12].

Radiotherapy is an essential treatment for anal canal tumors. Guidelines, however, do not differentiate between treatment volumes for perianal tumors and anal tumors. 

We conducted a literature review to analyze the sites at higher risk of harboring microscopic disease in patients with early-stage perianal cancer, to suggest a risk-adapted selection of RT elective volumes.

## 2. Materials and Methods

A comprehensive search for articles reporting on RT in pSCC was performed, using the search string reported in Appendix A. The PRISMA flowchart (Appendix B) was followed to transparently report on the manuscript selection process [13]. Given the rarity of the clinical scenario and the heterogeneous nature of the published literature, a narrative review was performed. Only manuscripts that included patients with early-stage anal margin tumors (T1–T2 N0 M0) treated with exclusive or postoperative RT were selected.

## 3. Results

### 3.1. Treatment

ESMO guidelines recommend concomitant 5 fluorouracil (5-FU) and mitomycin C (MMC) with intensity-modulated radiation therapy (IMRT) for early-stage pSCC [I, A] [14], based on the results of phase II and phase III randomized trials [EORTC 22861, United Kingdom Co-ordinating Committee on Cancer Research (UKCCCR) Anal Cancer Trial I (ACT I), RTOG 87-04, RTOG 98-11, ACCORD-03, RTOG 0529 and Cancer Research United Kingdom (CRUK) (ACT II)]. Nevertheless, early-stage true perianal tumors were poorly represented in those trials, so the applicability of the data to this clinical setting is limited. 

Data regarding local excision are mostly retrospective. Local excision is recommended for favorable lesions [II, A]. The latter are represented by T1 (<2 cm) or T2 tumors, provided a negative margin of at least 1 cm can be obtained without impairing the anal sphincter. A wide local excision can constitute an adequate treatment for these tumors because it preserves continence and achieves a good rate of local control [2]. This subgroup of patients can be treated with equal success with RT [2]. 

Resection margins are often involved or in close proximity and perianal SCC can recur in up to 40% of patients after local resection [15,16]. These patients need subsequent RT. In a series of 45 patients, predominantly with T1–2 anal margin SCCs treated with surgery (mostly without clear margins), 65% underwent local excision followed by RT, achieving a 5-year disease-free survival (DFS) rate of 86% [9]. Twenty-six patients, primarily with T2 tumors treated with local excision and RT, or RT alone, had a 5-year survival rate of 88.3% [10]. Historically, pSCCs were routinely treated with repeated excisions (as with resection of cutaneous SCCs of other body regions). Yet, the perianal anatomy hampers the usefulness of this strategy, since repeated excisions can result in significant morbidity. By contrast, as HPV-related anal cancers are extremely radiosensitive, lower RT doses are likely to achieve excellent outcomes for residual microscopic disease, with less toxicity than the higher doses required to treat skin cancers.

In clinical series, approximately 20–40% of patients have a histologically positive margin after local excision [14,17]. One possible reason is that excision often occurs before having a pathology assessment, as pSCCs can resemble numerous benign conditions. In addition, reconstruction of large defects in the perineal region is challenging.

Re-excision for perianal SCCs is discouraged in the ESMO guidelines, as it is associated with high morbidity, low histological yield, and long-term high local recurrence rates [14].

Previous ESMO–ESSO–ESTRO guidelines suggested that chemo-radiotherapy (CRT) could be used for pSCCs, both as a primary treatment or adjuvant therapy after local excision when the histology was poorly differentiated, irrespective of resection margin status [18]. The latest version of the ESMO guidelines established that no recommendation should be made based on tumor differentiation.

Conversely, NCCN guidelines [19] recommend CRT for poorly differentiated pSCCs, irrespective of T-stage, and they propose re-excision if the margins are not adequate, as commonly observed in skin cancers. In this regard, a retrospective cohort study from the National Cancer Database that included 2243 adults diagnosed with T1N0 anal canal cancer between 2004 and 2012 found that the use of local excision in this population increased over time (17.3% in 2004 to 30.8% in 2012; *p* < 0.001). There was no significant difference in the 5-year overall survival (OS) rate based on the management strategy (85.3% for local excision; 86.8% for chemo-radiotherapy; *p* = 0.93) [20]. 

Some ongoing studies are evaluating the optimal RT dose in anal and perianal carcinomas, aiming at de-escalating the dose in early-stage tumors, given their good radiosensitivity [21]. The ongoing PLATO (PersonaLising Anal cancer radioTherapy dOse, ISRCTN88455282) trial is a single protocol ‘umbrella platform’ comprising ACT3, 4, and 5 trials [22]. The PLATO ACT3 trial (*n* = 90) is a non-randomized phase II study evaluating a strategy of local excision for T1N0 anal margin tumors with selective postoperative involved field CRT. It prescribes 41.4 Gy to GTV in 23 fractions and concurrent capecitabine reserved for patients with margins ≤ 1 mm. PLATO ACT4 (*n* = 162) is a phase II randomized trial (2:1) comparing reduced and conventional dose RT (41.4 Gy in 23 fractions to GTV vs. CRT 50.4 Gy in 28 fractions) administered with IMRT and elective nodal irradiation to the pelvis and inguinal nodes, adding concurrent capecitabine for T1–2 (<4 cm) N0 disease [22].

The randomized phase II DECREASE study (NCT04166318) is currently testing lower-dose CRT compared to standard-dose CRT for patients with stage I or IIA anal or perianal cancer. Elective lymph node volumes are irradiated in all these studies, apart from the PLATO ACT3 trial.

### 3.2. Radiotherapy Volumes

As anal and perineal tumors are predominantly treated with local definitive treatments, pathological staging is absent, and cannot drive RT treatment volume selection. 

The use of IMRT relies on accurate target volume selection and delineation, which requires precise knowledge of the pattern of disease failure, for which data are very limited, especially regarding pSCCs.

ESMO guidelines suggest that RT treatment volumes initially encompass the primary tumor, anal canal, nodal regions, and inguinal nodes, with field reduction recommended to treat the primary tumor and sites of likely nodal involvement within the high-dose volume. ESMO guidelines refer to the Australasian Gastrointestinal Trials Group (AGITG) as a contouring atlas [14]. 

Currently, no study has considered modulating treatment volumes according to the stage of disease, creating a contradictory situation. In fact, early-stage pSCCs (T1–T2 N0 M0) that undergo local excision with clear margins can omit further treatments or undergo combined chemoradiotherapy with elective irradiation of the perineal region, inguinal lymph nodes and the pelvis. 

The lymphatic drainage of the anal margin is not solely directed to the inguinal lymph nodes. Dye injection studies have shown that it can proceed along the inferior rectal, middle rectal, superior hemorrhoidal, and inferior mesenteric vessels [23]. Perineal skin drainage, however, has not been extensively studied yet.

In the pelvic floor, superficial parietal vessels pass under the perineal skin from the coccygeal region up to the pubis. Anteriorly, they cross the medial side of the proximal thigh around the outer surface of the adductor muscles. Then, they join the supero-medial group of superficial inguinal lymph nodes. Their functional territory comprises all the soft tissues of the perineum, including those below the outer fascial sheath of the urogenital diaphragm, the inferior part of the anal canal below the ano-cutaneous line, and the caudal part of the vagina below the hymen.

Lymphatic spread for tumors of the perineal skin of the anal canal and vulva can occur through the superficial perineal accessory pelvic pathway. It is a subcutaneous pathway located anteriorly which joins the common terminal pelvic route of the external and common iliac lymphatic chains to the deep inguinal lymph nodes, ending in the superficial inguinal nodes. There are two different ways of dissemination to the external iliac nodes. The lymphatic spread of cancer cells directly from the primary tumor site is the first way. The second possibility is the dissemination of cancer cells from metastatic lymph nodes in the ipsilateral inguinal or, less frequently, from the ipsilateral internal iliac district [24]. 

Intensity-modulated radiation therapy or volumetric modulated arc therapy (VMAT) is recommended for the treatment of anal cancer in both the ESMO and NCCN guidelines [14,19]. 

Recently, ECOG ACRIN published guidelines to provide customized radiation treatment for early-stage (T1–2 N0 M0) patients with SCC of the anal canal and/or perianal skin treated with IMRT [25]. No distinction is made between anal canal and true perianal tumors.

The recommended clinical elective nodal CTV (CTVn) includes areas at risk for microscopic regional metastasis, embedding the external and internal iliac, inguinal, obturator, presacral, and mesorectal regions. The cranial border of CTVn is assessed at the level of the bifurcation of common iliac vessel into the external and internal iliac vessels, approximately located at the L5–S1 interspace.

Currently, pelvic magnetic resonance imaging (MRI) and fluorodeoxyglucose positron emission tomography (FDG-PET) are used to better define tumor (T) and nodal (N) stages. These new imaging technologies have limitations in terms of specificity for nodal involvement [26]. Nevertheless, they offer greater opportunities to identify subgroups requiring stratified treatment approaches.

In this regard, the Nordic anal cancer (NOAC) group recently published a consensus guideline for risk-adapted delineation of the elective clinical target volume in anal cancer [27]. This guideline did not separately analyze perineal tumors. They recommended that the elective nodal volume include the internal iliac, presacral, mesorectal, superior rectal, and inguinal regions and the entire anal canal.

Few retrospective series specifically analyzed the clinical results of early-stage perianal tumors treated with RT, as reported further in Table 2 [2,11,12,28,29,30,31].

#### 3.2.1. Inguinal Nodes

Regarding the inclusion of inguinal nodes in the elective treatment volumes for pSCCs, Ortholan et al. [32] reported a series of 181 patients with anal canal tumors and uninvolved inguinal nodes. The decision to treat the inguinal lymph nodes was at the physician’s discretion: 75 received elective inguinal irradiation (45 Gy in 25 fractions), while 106 only had the pelvic nodes treated. For T1–T2, the 5-year cumulative rate of inguinal recurrence without prophylactic irradiation was 12 vs. 3% in patients who received prophylactic inguinal treatment. Only 40 patients had anal margin involvement. 

In the TROG 99-02 study, 40 patients with T1–2 N0 anal carcinoma received CRT without inguinal node elective irradiation. The study was closed prematurely because of an unacceptable inguinal node relapse rate, as high as 22.5% at 3 years [33]. 

In another series with 283 patients who underwent definitive CRT for anal SCCs, two-thirds of inguinal failures occurred in node-negative patients, at a time when elective inguinal irradiation was not routine. In T1 N0 disease, the inguinal failure rate without elective irradiation was 1.9%, while it was 12.5% in T2 N0. On the other hand, there were no inguinal failures in patients who received elective nodal irradiation [34]. 

In a study by Zilli et al. considering 116 patients with T2 N0 anal cancer, isolated inguinal recurrence occurred in two patients (4.7%) treated without inguinal irradiation, whereas no groin relapse was observed in those whose treatment included inguinal nodes. Only 28% of these patients had perianal tumors or perianal skin involvement [35].

Patients with true perianal cancer, representing a minority of cases, were not separately analyzed in these studies. Staging relied on clinical examination or ultrasound and CT scan and did not include MRI or 18FDG-PET. Nevertheless, the finding of a 12–22.5% rate of inguinal lymph node recurrence suggests a higher risk of recurrence for true perianal tumors if inguinal nodes are not included in the irradiated volume. In fact, the inguinal lymph nodes are the first draining level of the perineal skin. Nevertheless, some reports show favorable results omitting inguinal irradiation for anal SCCs < 1 cm [36]. It should be investigated if the same results apply to pSCCs. The ACT3 study will provide useful information on the role of inguinal lymph node irradiation in pT1 tumors treated with postoperative RT [22].

NOAC guidelines consider unnecessary to include the posterolateral inguinal area in the elective CTV (CTVe), as three studies have shown that no LN metastasis was located there [37,38,39]. As the posterolateral inguinal area drains the lower limb, this choice could be adequate for node negative pSCCs. An example of inguinal node delineation can be found in Figure 2. 

#### 3.2.2. Ano-Inguinal Lymphatic Vessels

The ano-inguinal lymphatic drainage (AILD) comprises the subcutaneous adipose tissue of the proximal medial thigh and could be a risk volume to be included during elective irradiation for perianal tumors. There are no recommendations for including the ‘true’ AILD into the clinical target volume (CTV) in this setting. Dosimetric studies of anal cancer irradiated using IMRT have shown that only 76% of the AILD volume incidentally received at least the expected required treatment dose of 30 Gy [40]. In a retrospective series of one hundred and seventy anal canal patients, the overall rate of AILD recurrence was 1.2% (*n* = 2), and among patients with inguinal metastases at initial diagnosis, it was two out of sixty-five (3.1%). Only five patients had true pSCC in this series, making it questionable if, in this situation, AILD is at such a low risk of involvement that it can be spared. Therefore, taking the data into account, the likelihood of the hypothesis that there may be microscopic disease leading to recurrence in the AILD is low (risk < 3.1%). However, it should be highlighted that only a few pSCC patients were included in the available series.

Therefore, we do not recommend including the ‘true’ AILD in the contouring volume for every pSCC, but we must keep in mind that the inclusion of this anatomical region remains to be investigated and considered on a case-by-case basis. An example of AILD delineation can be found in Figure 3.

#### 3.2.3. Superior Pelvic Border

Considering the pelvic elective volume, in a retrospective mono-institutional analysis of 284 patients with anal canal tumors, the superior limit was consistently 1 cm above the sacroiliac inferior joints (SiIJ) or 5 cm proximal to the primary tumor, whichever was more proximal. Only nine patients (3.2%) experienced failure in the pelvic nodes, and three had simultaneous distant failure [34]. For patients with T1–2 N0 tumors not extending into the rectum, NOAC guidelines recommend that the cranial limit of the CTVe should be the inferior aspect of the sacroiliac joint, based on the results of two large studies [41,42,43].

In Norway, these low-risk patients have been treated with the cranial limit at the inferior face of the sacroiliac joint for many years, demonstrating that it is safe to lower the cranial border for low-risk patients.

If pelvic nodal inclusion is decided, maintaining the cranial border at the inferior face of the sacroiliac joint, instead of including internal iliac nodes until the bifurcation of the common iliac artery, could be a way to reduce overtreatment and toxicity. Figure 4 shows the two delineation options for the superior pelvic border. 

#### 3.2.4. External Iliac Level

In a series of 166 anal canal patients, all studied with PET-CT, 7.2% had positive external iliac and 7.2% internal iliac nodes [43]. 

NOAC consensus guidelines propose two equally accepted approaches: ‘Alternative A’ is to include the external iliac region for all patients. ‘Alternative B’ is to omit the external iliac region for patients with T1–2 N0 anal tumors [27].

In a recent analysis to investigate the patterns of PET-positive LNs in anal cancer, Frennered et al. [37] retrospectively assessed the baseline PET-CT status of 103 consecutive anal cancer patients. There were no patients with pSCCs. Still, a distinct pattern of nodal involvement was shown with 97% inguinal, 6% perirectal, 3% internal iliac, 24% external iliac, 0% common iliac, and 3% paraaortic nodal involvement in 33 patients with tumors of the anal canal extending to the perianal skin versus 60% inguinal, 30% perirectal, 10% internal iliac, 30% external iliac, 30% common iliac, and 10% paraortic nodal involvement in 10 patients with anal tumors without perineal skin extension. Cases of anal canal tumors with perianal extension were more likely to have inguinal solitary region localization (94%, sixteen out of seventeen), while patients with anal canal tumors with extension into the rectum were more likely to have perirectal solitary region localization (60%, nine out of fifteen).

Therefore, the external iliac node station should remain as a part of the adjuvant pelvic lymph nodes, at least for patients with larger tumors (T2 or more).

#### 3.2.5. Pararectal, Presacral, and Internal Iliac Nodes

Pararectal lymph nodes are located around the rectum, posteriorly to the digestive pelvic compartment. These nodes are reached by vessels arising directly from the neighboring viscera. On the other hand, efferent vessels extend to the internal iliac, external iliac, or presacral chains [24]. These volumes could probably be omitted for patients with T1–T2 N0 pSCCs not involving the anal canal, although no data are available to support this suggestion.

## 4. Suggested Radiotherapy Elective Volumes in Perianal Squamous Cell Carcinomas

Few retrospective series specifically analyzed the clinical results for early-stage perianal tumors treated with RT. Table 2 summarizes these results.

It should be noted that these studies were performed in years when early-stage tumors of the anal margin were irradiated as skin tumors with mixed techniques, such as interstitial brachytherapy and direct electron or photon fields. Therefore, elective irradiation was usually limited to inguinal chains irradiated with electrons in most cases, and rarely extended to pelvic and inguinal levels with a box technique. Among the 196 patients reported in these series of studies, there were only eight pelvic nodal recurrences, none of which were isolated or limited to pelvic lymph nodes. However, the retrospective nature of these studies, the small number of patients, and the heterogeneity of the treatment schedules over a long period make it rather difficult to establish conclusions about the role of prophylactic lymph nodal volume. Considering all of these, we hereby provide suggestions on volumes to be irradiated in this setting.

In T1 or T2, N0 treatment volumes should encompass the primary tumor. Gross tumor volume (GTV) delineation should be based on clinical assessment, anorectal endoscopy, contrast-enhanced CT, pelvic magnetic resonance imaging (MRI), and FDG PET/CT. The GTV should be expanded isotropically (20 mm) to generate the primary clinical target volume (CTV) and edited to exclude bone and muscles. The anal complex should be included in the primary tumor CTV. A 5 to 10 mm planning target volume (PTV) margin is suggested based on institutional setup and patient-specific factors.

The elective volume should encompass: The perineal skin and the superficial perineal accessory pelvic pathway (all the soft tissues of the perineum comprising those below the outer fascial sheath of the urogenital diaphragm, the inferior part of canal anal below the ano-cutaneous line, and the caudal part of the vagina below the hymen) should be included.The ano-inguinal lymphatic drainage (AILD) that is located in the subcutaneous adipose tissue of the proximal medial thigh can be considered as a risk volume. We do not recommend the routine inclusion of the ‘true’ AILD in the contouring volume for pSCCs, but we highlight the fact that the management of this anatomical region remains an issue to be investigated and considered on a case-by-case basis.We suggest including the inguinal nodes within the elective volumes. Nevertheless, the omission of inguinal irradiation for SCCs of anal canal cancers < 1 cm could be an option.The external iliac node station should remain as a part of the adjuvant treatment, at least for patients with larger tumors (T2 or bigger).

In well-differentiated T1 tumors, the elective irradiation of inguinal and external iliac nodes may be omitted. The pararectal, presacral, and internal iliac nodes could probably be omitted for patients with T1–T2 N0 pSCCs not involving the anal canal. If pelvic nodal inclusion is decided, maintaining the cranial limit at the inferior face of the sacroiliac joint, instead of including internal iliac nodes until the bifurcation of the common iliac artery, could be a way to reduce overtreatment and toxicity.

Table 3 below summarizes suggested radiotherapy elective volumes in perianal squamous cell carcinomas.

Additional reports of nodal level distribution with PET-CT at diagnosis, patterns of failure following definitive RT for perineal cancer, and detailed anatomic studies will provide information to help refine target volume selection. Given the rarity of this condition, prospective registry studies are necessary.

## 5. Conclusions

Being aware that the discussed data are mostly retrospective and hence not consistent, a few suggestions can be inferred. The irradiated elective volume may be decreased for small pSCC N0 tumors, potentially decreasing morbidity and toxicity [44]. Since recurrences are difficult to manage, it seems that elective pelvic lymph node irradiation (including the external iliac level) should be recommended for patients with large T2 tumors with negative lymph nodes, although a reduction in the cranial limit could be proposed. On the other hand, T1 and small T2 cases could probably take advantage of smaller treatment volumes, as proposed according to the data available to date and waiting for those coming from new perspective studies.

## Figures and Tables

**Figure 1 cancers-15-05833-f001:**
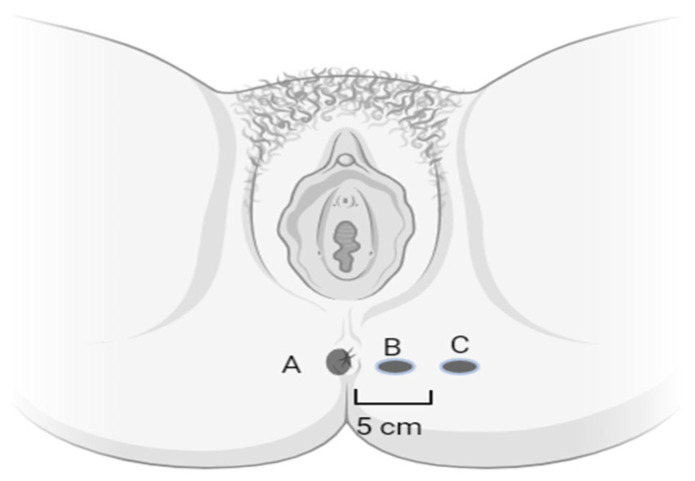
Anal canal (A), perianal skin (B), and skin cancer (C).

**Figure 2 cancers-15-05833-f002:**
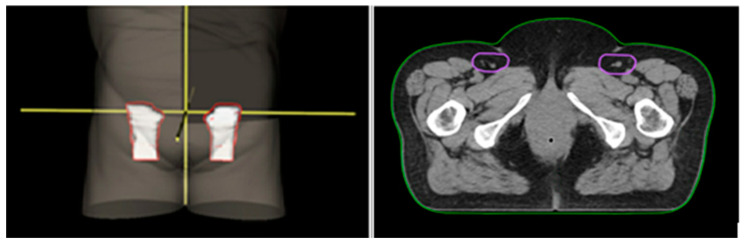
Inguinal node delineation.

**Figure 3 cancers-15-05833-f003:**
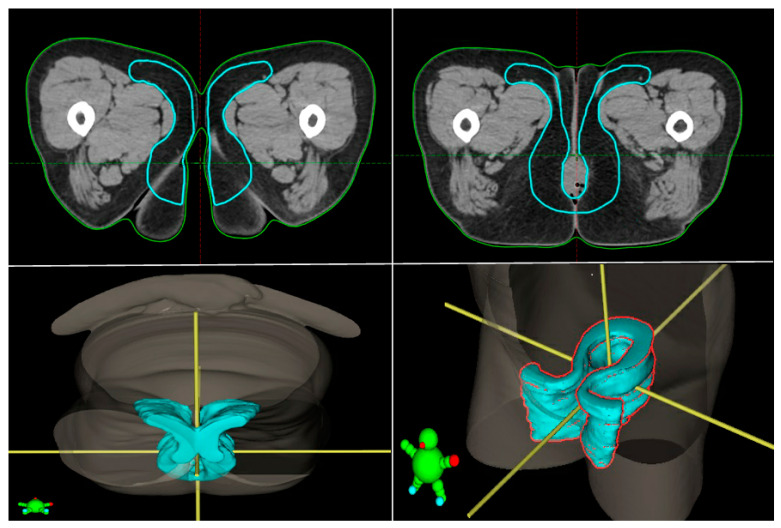
Ano-inguinal lymphatic drainage (AILD) delineation.

**Figure 4 cancers-15-05833-f004:**
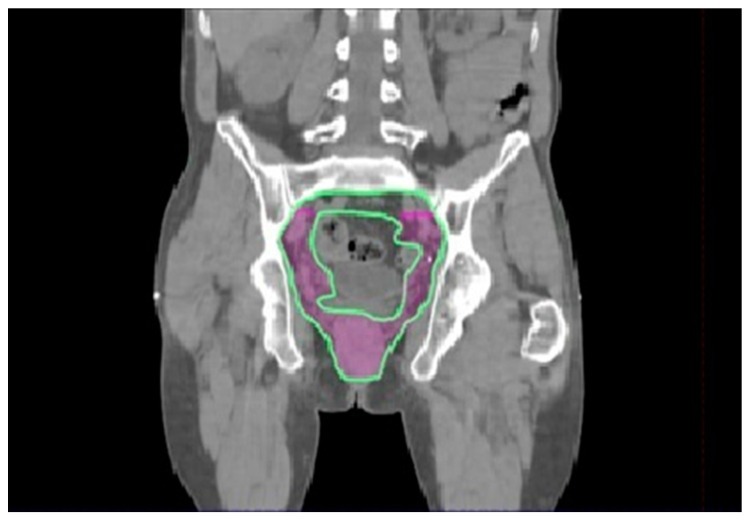
Cranial border of the pelvic elective volume: in violet, the cranial border at the SiIJ is shown; in green, the superior limit is shown, including the internal iliac nodes until the bifurcation of the common iliac artery.

**Table 1 cancers-15-05833-t001:** TNM classifications for anal cancer.

**Primary Tumor (T)**	
Tx	Primary tumor cannot be assessed
T0	No evidence of primary tumor
Tis	High-grade squamous intraepithelial lesion (previously termed carcinoma in situ, Bowen disease, anal intraepithelial neoplasia II–III, or high-grade anal intraepithelial neoplasia)
T1	Tumor ≤ 2 cm in greatest dimension
T2	Tumor > 2 cm but ≤5 cm in greatest dimension
T3	Tumor > 5 cm in greatest dimension
T4	Tumor of any size invades adjacent organ(s) (e.g., vagina, urethra, bladder)
**Regional lymph nodes** **(N)**	
Nx	Regional lymph nodes cannot be assessed
N0	No regional lymph node metastasis
N1	Metastasis in inguinal, mesorectal, internal iliac, or external iliac nodes
	N1a—Metastasis in inguinal, mesorectal, or internal iliac lymph nodes
	N1b—Metastasis in external iliac lymph nodes
	N1c—Metastasis in external iliac with any N1a nodes
**Distant Metastasis** **(M)**	
M0	No distant metastasis
M1	Distant metastasis

**Table 2 cancers-15-05833-t002:** Clinical results of retrospective studies for early-stage perianal tumors treated with RT.

Study	N° of Patients	Stage	Elective Volume	Total Dose Gy	Concomitant CHT	Inguinal Relapse	Pelvic Nodal Recurrence	LF	DSS	OS	Fecal Inconti- nence	Anal Sphincter Preservation Rate
**Balamucki, 2011** [11]	26	Mostly T2 N0	External iliac and inguinal LNs, except for T > 2 where the pelvis is included	59	19% T2, 100% T3	9% (pts not irradiated on inguinal LN)	3.8% (only one pt, with AIDS)	9%	92% 10 y	56% 10 y	12%	88%
**Newlin,** **2004** [2]	19	Mostly T2 N0	The external iliac LNs and inguinal LNs, except for T > 3 where the pelvis is included	50–65	5% T2, 100% T3	5% (1 pt not irradiated in inguinal LN)	0%	0%	95% 10 y	74% 10 y	21%	N/A
**Papillon, 1992** [12]	57	Mostly T2 N0	No prophylactic nodal RT	60	only T3	16%	N/A	12.5%	81.5% 5 y	59.2% 5 y	N/A	N/A
**Peiffert, 1997** [28]	32	Mostly T2 N0	No prophylactic nodal RT	60.5	No	8%	N/A	23% 5 y	89% 5 y	65% 5 y	N/A	84%
**Cutuli, 1988** [29]	21	50% T1–T2 N0	Inguinal nodes 50%	65	No	15% (none received inguinal prophylactic irradiation)	N/A	N/A	62% 5 y	52% 5 y	N/A	90%
**Toubul, 1995** [30]	17	Mostly T2 N0	Pelvic + inguinal nodes 41%	65	No	8%	N/A	13.5% 5 y	72% 5 y	48% 5 y	N/A	82% 5y
**Bieri,** **2001** [31]	24	Mostly T2 N0	58% pelvic + inguinal, 70% inguinal	57.8	58%	0%	0%	29% 5 y	69.5% 5 y	56% 5 y	4%	N/A

Abbreviations: CHT: chemotherapy; LF: local failure, to be intended as failure within the primary tumor site; DSS: disease-specific survival; OS: overall survival; LN: lymph nodes; Y: years; N/A: not available; Pt: patient; AIDS: acquired immune deficiency syndrome.

**Table 3 cancers-15-05833-t003:** Suggested radiotherapy elective volumes in perianal squamous cell carcinomas.

	Perineal Skin and Superficial Perineal Accessory Pelvic Pathway	Ano-Inguinal Lymphatic Vessels (AILD)	Inguinal Nodes	External Iliac Level	Pararectal and Presacral and Internal Iliac Nodes
**T1 N0**	Yes	No	Yes, may be omitted in pSCCs <1 cm	May be omitted	May be omitted * ++
**T2 N0 ‘small’** **(<4 cm)**	Yes	To be considered (<5% risk of recurrence in anal canal series)	Yes	Yes	May be omitted * ++
**T2 N0 ‘large’**	Yes	To be considered (<5% risk of recurrence in anal canal series)	Yes	Yes	Yes, maintaining the cranial limit at the inferior face of SiIJ could be considered

* pSCCs not involving the anal canal. ++ If pelvic nodal inclusion is decided, maintaining the cranial limit at the inferior face of sacroiliac joint could be considered.

## Data Availability

The data presented in this study are available on request from the corresponding author. The data are not publicly available due to privacy reasons.

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
