# Peer review of "Nodal Elective Volume Selection and Definition during Radiation Therapy for Early Stage (T1–T2 N0 M0) Perianal Squamous Cell Carcinoma: A Narrative Clinical Review and Critical Appraisal"

_cancers, 2023, doi:10.3390/cancers15245833_

Round 1

Reviewer 1 Report

Comments and Suggestions for Authors

I want to thank the Authors for the interesting review sent for publication in Cancers.

It is of great interest and I recommend the publication after clarifying the following comments.

Appendix A shows the search strategy, and Appendix B the PRISMA flowchart. Only papers that included patients with early-stage anal margin tumors (T1-T2 N0 M0) treated with exclusive or postoperative RT were selected.

The Authors perform a PRISMA chart, they want to perform a systematic review? If yes I would suggest to better define inclusion or exclusion criteria. In the title the Authors state that their review is a narrative one and not a systematic review, in such a case a PRISMA is not necessary.

Results: the majority of the studies /guidelines cited are not part of the PRISMA, I think it should be explain in M&M for clearness.

However, I suggest in result section to move in paragraph 4 the results of PRISMA /table and suggested lymphnodes for perianal.

Introduction: define the aim of the review (it is partially define in M&M, move in introduction)

Minor:

Typos: line 135 “… ore…” correct with “are”

Line 270: check reference I suppose is number 35 and not 34.

Check the number reference in particular those referred to table 2

Author Response

I want to thank the Authors for the interesting review sent for publication in Cancers.

It is of great interest and I recommend the publication after clarifying the following comments.

Appendix A shows the search strategy, and Appendix B the PRISMA flowchart. Only papers that included patients with early-stage anal margin tumors (T1-T2 N0 M0) treated with exclusive or postoperative RT were selected.

The Authors perform a PRISMA chart, they want to perform a systematic review? If yes I would suggest to better define inclusion or exclusion criteria. In the title the Authors state that their review is a narrative one and not a systematic review, in such a case a PRISMA is not necessary.

Thank you for the comment. Given the paucity of data and the heterogeneous nature of the literature regarding the topic, we decided to perform a narrative review. A systematic review was not considered suitable in this context. However, we decided to follow the PRISMA flowchart to report in a transparent and clear manner the search strategy that was employed. We conducted a search based on the search string that we reported in the appendix A and we decided to add the PRISMA diagram to improve consistency in reporting. We modified the text and defined out review comprehensive and narrative (but not systematic).

Results: the majority of the studies /guidelines cited are not part of the PRISMA, I think it should be explain in M&M for clearness.

We revised the Prisma diagram to include all the manuscripts on which we focused in our review.

However, I suggest in result section to move in paragraph 4 the results of PRISMA /table and suggested lymphnodes for perianal.

 Thank you for your suggestion. We moved it to paragraph 4, with a link to the text.

Introduction: define the aim of the review (it is partially define in M&M, move in introduction)

The aim was defined in the introduction.

Minor:

Typos: line 135 “… ore…” correct with “are”.  

Ok thank you. Modified accordingly.

Line 270: check reference I suppose is number 35 and not 34.

 Ok thank you. Modified accordingly.

Check the number reference in particular those referred to table 2.

Ok thank you. This has been checked.

Reviewer 2 Report

Comments and Suggestions for Authors

This manuscript deals with the radiotherapy target volume for rare early-stage perianal cancers. To suggest elective treatment volumes, the authors reviewed many articles, including contouring guidelines, prospective study protocols, and case series with radiotherapy. Although the authors proposed elective treatment volumes at the end of the manuscript, supporting evidence provided by the authors seems insufficient. When the rate of recurrence is low in the superficial perineal accessory pelvic pathway and the ano-inguinal lymphatic drainage, the suggested wide elective volumes could have the risk of overtreatment. Furthermore, the authors may need to review the patterns of failure after surgical treatment alone. Finally, I recommend that the authors revise the suggestions according to the magnitude of recurrence risk in the elective volumes, which should consider the risks and benefits of wide elective treatment volumes.

Detailed comments:

-      Line 192: The period between nodes and with should be replaced with a comma.

-      Table 2 needs the definitions for the abbreviations.

-      Line 251: Even though only 8 regional recurrences occurred among the 196 patients of the studies shown in Table 2, Table 2 shows higher rates of inguinal relapse. More detailed and specific explanations are needed (what is the definition of regional recurrence?). In addition, elective regional irradiation may be unnecessary in early-stage perianal cancer if only 8 (4%) out of 196 patients experienced regional relapse. Furthermore, all of these patients experienced other recurrences in addition to isolated or limited pelvic LNs.

-      Figure 2 did not delineate the inguinal node according to the NOAC consensus. The authors should choose the CT slice and delineate the CTV as demonstrated in Figure 2 (6th row and 2nd column) of the NOAC consensus to align with what is described in the manuscript..

-      Line 306: What does the following sentence mean? Does it imply that there is inadequate evidence to treat AILD? Or is a wider margin of greater than 2.5 cm needed to fully cover the AILD?
“The indication to include in pCTV 2.5 cm around GTV could be suboptimal for the adequate covering of this volume.”

-      Line 313: The expression 'sacroiliac inferior joint' should be revised, as the acronym 'SIJ' refers to 'sacroiliac joint' in the rest of the manuscript.

-      Section 3.2.4 External iliac level: There is no supporting data to guide the delineation of the target volume for early-stage perianal cancer. The authors need to provide more evidence on whether to include the external iliac area electively.

-      Page 11: The authors’s suggestion should be based on evidence, including recurrence rate or impact on survival outcomes, as I already mentioned at the beginning of this review report. The contents listed under the second bullet point should be divided into two separate sections: one for inguinal nodes and the other for AILD. In addition, why is a 10mm margin necessary to create the CTV?

-      Line 393: The citation of reference 45 (self-citation) is not suitable, as it does not contain relevant issues.

-      Citation numbers require correction as there are mismatches and instances of duplication.

Author Response

This manuscript deals with the radiotherapy target volume for rare early-stage perianal cancers. To suggest elective treatment volumes, the authors reviewed many articles, including contouring guidelines, prospective study protocols, and case series with radiotherapy. Although the authors proposed elective treatment volumes at the end of the manuscript, supporting evidence provided by the authors seems insufficient. When the rate of recurrence is low in the superficial perineal accessory pelvic pathway and the ano-inguinal lymphatic drainage, the suggested wide elective volumes could have the risk of overtreatment. Furthermore, the authors may need to review the patterns of failure after surgical treatment alone. Finally, I recommend that the authors revise the suggestions according to the magnitude of recurrence risk in the elective volumes, which should consider the risks and benefits of wide elective treatment volumes.

Thank you for your careful analysis of the paper. Regarding the review of surgical case series in this setting, they are few and very fragmentary. They are limited to early-stage tumors and do not help us in the choice of elective volumes delineation, particularly for T2 tumors. In fact, surgical series are very heterogeneous, with many re-operations and inconsistent follow-up.  Therefore, defining pattern of failure after surgical treatment alone is not very significant, unlike other tumors (e.i. head and neck).

Detailed comments:

-      Line 192: The period between nodes and with should be replaced with a comma.

        Ok, thank you. It has been replaced.

-----------------------------

-      Table 2 needs the definitions for the abbreviations.

Thank you, we added the abbreviations to the table.

-------------------------------

-      Line 251: Even though only 8 regional recurrences occurred among the 196 patients of the studies shown in Table 2, Table 2 shows higher rates of inguinal relapse. More detailed and specific explanations are needed (what is the definition of regional recurrence?). In addition, elective regional irradiation may be unnecessary in early-stage perianal cancer if only 8 (4%) out of 196 patients experienced regional relapse. Furthermore, all of these patients experienced other recurrences in addition to isolated or limited pelvic LNs.

Thank you for your analysis. The table includes only inguinal lymph node recurrences (inguinal relapse) and T-site recurrences (local failure, LF). Data (albeit with some difficulty in interpretation, given the heterogeneous attitude on prophylactic irradiation of inguinal lymph nodes) show that inguinal lymph node recurrences can be as high as 15-16% if the inguinal level is not included in the elective volumes. We have modified the sentence as follows for clarity: “Among the 196 patients reported in these series of studies, there were 8 pelvic nodal recurrences, none of which were isolated or limited to pelvic lymph nodes”.

------------------------------------------------

-      Figure 2 did not delineate the inguinal node according to the NOAC consensus. The authors should choose the CT slice and delineate the CTV as demonstrated in Figure 2 (6th row and 2nd column) of the NOAC consensus to align with what is described in the manuscript.

We have modified it, thank you.

----------------------------

-      Line 306: What does the following sentence mean? Does it imply that there is inadequate evidence to treat AILD? Or is a wider margin of greater than 2.5 cm needed to fully cover the AILD?
“The indication to include in pCTV 2.5 cm around GTV could be suboptimal for the adequate covering of this volume.”

Thanks for the comment. We have modified the text in this way: “Therefore, taking the data into account, the hypothesis that there may be microscopic disease leading to recurrence in AILD is low (risk <3.1 %). However, we would like to highlight that in the series only few pSCC patients were included. Patterns of recurrence in anal cancer: a detailed analysis. Therefore, overall we do not recommend to include the ‘true’ AILD in the contouring volume for every pSCC, but we must keep in mind that this anatomical region remains an issue that needs to be investigated and to be considered on a case-by-case basis”.

The sentence: ‘indication to include in pCTV 2.5 cm around GTV could be suboptimal for the adequate covering of this volume’.

Thanks for the comment. We have removed this sentence from the text.

-------

-      Line 313: The expression 'sacroiliac inferior joint' should be revised, as the acronym 'SIJ' refers to 'sacroiliac joint' in the rest of the manuscript.

We modified the text as follow: “Considering the pelvic elective volume, in a retrospective mono-institutional analysis of 284 patients with anal canal tumors, the superior limit was consistently 1 cm above the sacroiliac inferior joints (SiIJ) or 5 cm proximal to the primary tumor, whichever was more proximal”. We have also changed the abbreviation in the text to make reading more accessible. Thank you very much for the useful clarification, now it is clear for the reader that these are different cranial limits.

---------------------

-      Section 3.2.4 External iliac level: There is no supporting data to guide the delineation of the target volume for early-stage perianal cancer. The authors need to provide more evidence on whether to include the external iliac area electively

The evidence is scant. Nevertheless, most of the guidelines to suggest the inclusion of external iliac nodes for early-stage cancer. Even the most permissive guidelines (NOAC), consider the option of inclusion. Upon a cautionary criterion, we suggested including the external iliac nodes.

-      Page 11: The authors’s suggestion should be based on evidence, including recurrence rate or impact on survival outcomes, as I already mentioned at the beginning of this review report. The contents listed under the second bullet point should be divided into two separate sections: one for inguinal nodes and the other for AILD. In addition, why is a 10mm margin necessary to create the CTV?

Thank you for the request. We have modified text as suggested in the previous comments. We also added to bulled point and to Table 3 our considerations. We decided to write the statements in the bullet without reporting the risk of recurrence, following the lead of other authors (e.i. Nordik guidelines in which the risk percentages are not reported). This idea is also justified by the fact that, as previously reported, there is very little evidence for perianal tumors, even in surgical case series. In fact, there are few retrospective case series, sometimes with reoperations, and recurrence analyses are not performed differentiating the lymph node levels. Therefore, defining pattern of failure after surgical treatment alone is not very significant, unlike other tumors (e.i. head and neck).

We have created two sections as rightly suggested.

The elective volume should encompass:

  • Perineal skin and the superficial perineal accessory pelvic pathway (all the soft tissues of the perineum below the outer fascial sheath of the urogenital diaphragm, but also the distal part of the vagina below the hymen and the inferior part of the anal canal below the ano-cutaneous line).
  • The ano-inguinal lymphatic drainage (AILD) that is located in the subcutaneous adipose tissue of the proximal medial thigh can be a risk volume. We do not recommend to include the ‘true’ AILD in the contouring volume for pSCC, but we must keep in mind that this anatomical region remains an issue that needs to be investigated and to be considered on a case-by-case basis.
  • Inguinal nodes. We suggest to include in elective volumes inguinal nodes. Omitting inguinal irradiation for SCC of anal canal cancers < 1 cm could be considered.
  • The external iliac node station should remain as a part of the adjuvant treatment, at least for patients with larger tumors (T2 or bigger).

In addition, why is a 10mm margin necessary to create the CTV?

We have modified our sentence as follows: “ The GTV should be expanded isotropically (20 mm) to generate the primary clinical target volume (CTV) and edited to exclude bone and muscles”. Thank you.

-      Line 393: The citation of reference 45 (self-citation) is not suitable, as it does not contain relevant issues. 

Ok, thank you. It has been removed.

-      Citation numbers require correction as there are mismatches and instances of duplication.

Ok, thank you. This has been corrected.

Round 2

Reviewer 2 Report

Comments and Suggestions for Authors

The paper has been well-revised according to the instructions. However, there are some mistakes in the manuscript.

- Line 210: “groin” should be replaced with “chains”

- Figure 2: The authors should select the CT image below the provided CT slice, similar to the image shown in Figure 2 (6th row and 2nd column) of the NOAC consensus.

- Table 2: The addition of the number of patients with pelvic nodal recurrence is recommended. Does ‘ex-T site’ mean failure in the primary site? Then, ‘ex-T site’ should be replaced with the definition of local failure. Check the journal style for “Abbreviations.”

- Line 358: 'inguinal groins' should be replaced with another term.

- Line 360: ‘8 regional recurrences’ should be replaced with ‘ 8 pelvic nodal recurrences.’

- Line 409: One ‘that’ should be deleted.

  Comments on the Quality of English Language

The authors need to compile sentences with similar content into one paragraph.

There are lots of sentences that are difficult to read.

Author Response

The paper has been well-revised according to the instructions. However, there are some mistakes in the manuscript.

Line 210: “groin” should be replaced with “chains”.

Thanks. This has been modified accordingly.

- Figure 2: The authors should select the CT image below the provided CT slice, similar to the image shown in Figure 2 (6th row and 2nd column) of the NOAC consensus.

Thanks. Figure 2 has been modified accordingly.

- Table 2: The addition of the number of patients with pelvic nodal recurrence is recommended. Does ‘ex-T site’ mean failure in the primary site? Then, ‘ex-T site’ should be replaced with the definition of local failure. Check the journal style for “Abbreviations.”

Thank you for your comment. We have updated and clarified the definitions. A column has been added to Table 2 named ‘pelvic nodal recurrence’. This latter data is not available in most studies because patients were staged with physical examination, chest x-ray and abdominal ultrasound.

- Line 358: 'inguinal groins' should be replaced with another term.

Thanks. This has been modified accordingly.

- Line 360: ‘8 regional recurrences’ should be replaced with ‘8 pelvic nodal recurrences.’

Thanks. This has been modified accordingly.

- Line 409: One ‘that’ should be deleted.

Thanks. This has been modified accordingly.

Comments on the Quality of English Language

The authors need to compile sentences with similar content into one paragraph.

There are lots of sentences that are difficult to read.

Thanks. The language has been simplified and (hopefully) optimized, as suggested.